# Sensor Fusion for Simultaneous Estimation of In-Plane Permeability and Porosity of Fiber Reinforcement in Resin Transfer Molding

**DOI:** 10.3390/polym14132652

**Published:** 2022-06-29

**Authors:** Wei Qi, Tzu-Heng Chiu, Yi-Kai Kao, Yuan Yao, Yu-Ho Chen, Hsun Yang, Chen-Chieh Wang, Chia-Hsiang Hsu, Rong-Yeu Chang

**Affiliations:** 1School of Information and Electrical Engineering, Zhejiang University City College, Hangzhou 310015, China; qiw@zucc.edu.cn; 2Department of Chemical Engineering, National Tsing Hua University, Hsinchu City 30013, Taiwan; e6060372@yahoo.com.tw (T.-H.C.); ss25686349@gmail.com (Y.-K.K.); 3CoreTech System Co., Ltd. Headquarters, Tai Yuen Hi-Tech Industrial Park, Chupei City, 30265, Taiwan; zoechen@moldex3d.com (Y.-H.C.); fredyang@moldex3d.com (H.Y.); jyewang@moldex3d.com (C.-C.W.); davidhsu@moldex3d.com (C.-H.H.); rychang@moldex3d.com (R.-Y.C.)

**Keywords:** polymer composites, resin transfer molding, permeability, porosity, measurement system, numerical simulation

## Abstract

To meet the expectation of the industry, resin transfer molding (RTM) has become one of the most promising polymer processing methods to manufacture fiber-reinforced plastics (FRPs) with light weight, high strength, and multifunctional features. The permeability and porosity of fiber reinforcements are two of the primary properties that control the flow of resin in fibers and are critical to numerical simulations of RTM. In the past, various permeability measurement methods have been developed in the literature. However, limitations still exist. Furthermore, porosity is often measured independently of permeability. As a result, the two measurements do not necessarily relate to the same entity, which may increase the time and labor costs associated with experiments and affect result interpretation. In this work, a measurement system was developed by fusing the signals from capacitive sensing and flow visualization, based on which a novel algorithm was developed. Without complicated sensor design or expensive instrumentation, both in-plane permeability and porosity can be simultaneously estimated. The feasibility of the proposed method was illustrated by experiments and verified with numerical simulations.

## 1. Introduction

In recent years, resin transfer molding (RTM) has become a promising manufacturing method to produce fiber-reinforced plastics (FRPs), which have been applied to a wide range of fields, such as civil engineering, automotive industry, shipbuilding, and aerospace industry, due to their light weight, high strength, and multifunctional features. During RTM, thermosetting resin is injected into a mold and impregnates the fiber preform placed in the mold cavity. After curing, the mold is opened to take out the composite product. 

In order to achieve better process understanding and product design, numerical simulation software is an essential tool providing numerical simulations of RTM [1,2,3]. In flow simulations, the permeability and porosity of fiber reinforcements are two important parameters, because they control the resin flow in fibers. In the past years, various permeability measurement methods have been proposed [4,5]. According to the saturation state of the fabric media, there are two types of permeability, i.e., unsaturated permeability and saturated permeability [6,7]. In RTM, the former is often considered to be more important. Usually, three permeability components are needed to fully characterize fluid flow in an anisotropic media, including two in-plane components and one out-of-plane component [8]. In the literature, most related studies talk about the measurement of in-plane permeability. In the following of this paper, we also focus on unsaturated in-plane permeability.

Darcy’s law is the core of permeability measurement, based on which the in-plane permeability can be inferred from the relationship between the flow rate at the flow front and the pressure drop. Visualization systems composed of CCD cameras and transparent mold plates are often used to obtain the position and velocity information of the flow front [9,10,11,12,13]. In recent years, dielectric sensors have been adopted in some research works for monitoring the resin flow [14,15,16,17]. In the experiments of permeability measurement, most existing methods take the porosity of the fiber preform as a known constant in the Darcy’s law. Usually, the value of porosity is independently measured before conducting the permeability measurement, which means that these two factors do not necessarily relate to the same entity. However, it is known that porosity and permeability are related properties [18]. Therefore, it is desirable to measure these two factors in a simultaneous manner.

In this work, a measurement system is developed by integrating capacitive sensing and resin flow visualization, where a parallel-plate capacitor is installed in a transparent mold made of poly(methyl methacrylate) (PMMA) and a CCD camera is used to capture the movement of the resin flow. The parallel-plate capacitor is not a new technique. However, to the best of our knowledge, it has not been used for porosity measurement in RTM. In addition, with the developed algorithms, both the in-plane permeability and porosity of the fiber preform placed inside the mold can be estimated simultaneously with a single experiment. No complicated sensor design or expansive instrumentation is required during the measurement experiments. These are the main contributions of this work.

The rest of this paper is organized as follows. In Section 2, the experimental setup is introduced, followed by the proposed measurement algorithms derived in Section 3. Then, experimental results are shown in Section 4 to illustrate the feasibility of the proposed method. In addition, the measurement accuracy was verified using numerical simulations. Finally, conclusions are made in Section 5.

## 2. Experimental Setup

The process diagram of the experimental system is plotted in Figure 1. The inlet of the transparent mold is connected to a resin bucket, while its outlet is linked to a vacuum pump that enables the impregnation of fiber preform with resin. As shown in Figure 2, the mold is made of PMMA, with a size of 40 cm × 20 cm × 0.9 cm. The size of the mold cavity containing the preform is 33 cm × 12 cm × 0.3 cm. Figure 3 is a conceptual illustration of the transparent mold, which is composed of three layers of PMMA plates: while the bottom plate is rectangular in shape, the center of the middle plate is hollow which forms the mold cavity, and the upper plate contains a slit-shaped inlet and outlet. The mold is formed by stacking these three plates together. The usage of the slit-shaped inlet and outlet relieves the race-tracking effect, which often occurs during the resin infusion. 

A parallel-plate capacitor composed of two 1.5-cm-wide copper foil tapes is installed inside the mold cavity, while the strips stick to the top and bottom plate of the mold, respectively. As shown in Figure 4, a circuit board is used for measuring the capacitance values, which uses the RS-232 serial protocol for communication. The capacitance measuring circuit was designed based on AD7746 from Analog Devices, where AD7746 is a high-resolution capacitance-to-digital converter with a resolution down to 4 aF and an accuracy of 4 fF. By combining the programmable on-chip digital-to-capacitance converter (CAPDAC) and the range extension circuit, the measuring range is extended to 230 pF.

The data acquisition is achieved by a computer with a LabVIEW program designed and coded in-house. A high-speed CCD camera and a National Instruments (NI) IMAQ frame grabber card installed in the computer are used to capture the flow front positions in real time. 

The preform inside the mold cavity is made of glass fiber sheets displayed in Figure 5. Before conducting the resin infusion, the fiber sheet was cut to fit the shape of the mold cavity. Then, sealant tape was used to seal the edges of the fiber sheet inside the mold. As shown in Figure 2, the black sealant tape can be observed around the inside circumference of the mold cavity. In doing this, the effect of race-tracking is further relieved.

The resin used in this work was epoxy resin produced by Swancor Ind. Co., Ltd. (Nantou City, Taiwan), of the type 2502-A. The dielectric constant of the resin was measured as 4.25. 

## 3. Methodologies

### 3.1. Capacitance Sensing for Porosity Measurement

As introduced in the previous section, a parallel-plate capacitor is installed inside the mold cavity to sense the flow of the resin. It is known that the capacitance between two parallel plates is defined as
(1)$$C=\varepsilon \varepsilon_{0}\;\frac{A}{d},\;$$
where *C* is the capacitance, *A* is the plate area, d is the distance between two plates, ε is the dielectric constant of the medium, and $\varepsilon_{0}=8.855\;\mathrm{pF}/m$ is vacuum permittivity. In the RTM experiments, the values of the area *A* and distance d are constants. Therefore, the change in capacitance is only attributed to the variation of the dielectric constant ε. 

As illustrated in Figure 6, at any time during the infusion of resin, the mold cavity is separated into two regions by the flow front, where the fibers in Region 1 have been infused by resin and those in Region 2 are still unoccupied. According to the Lichtenecker’s equation [19], the mixed dielectric constant $\varepsilon_{mix}$ of the two materials was calculated as
(2)$${log(\varepsilon }_{mix}{)=V}_{1}\;log\;\left(\varepsilon_{1}\right){+V}_{2}\;log\;\left(\varepsilon_{2}\right),\;$$
where $\varepsilon_{1}$ and $\varepsilon_{2}$ are the dielectric constants of two materials, respectively, and *V*_1_ and *V*_2_ are the corresponding volume ratios. The medium in Region 1 is a mixture of fibers and resin. Therefore,
(3)$${log(\varepsilon }_{mix1}{)=V}_{f1}\;log\;\left(\varepsilon_{f1}\right){+V}_{r}log\;\left(\varepsilon_{r}\right),\;$$
where $\varepsilon_{mix1}$ is the mixed dielectric constant of the medium between the copper foil tapes in Region 1; ${\;V}_{f1}$ and $\varepsilon_{f1}$ are the volume ratio and dielectric constant of the fibers in this region, respectively; and $V_{r}$ and $\varepsilon_{r}$ are defined in a similar way for the resin. From (3), it is easy to derive that
(4)$$\varepsilon_{mix1}=\varepsilon_{f1}{}^{V_{f1}}\varepsilon_{r}{}^{V_{r}}\;$$

Similarly, in Region 2, we have
(5)$$\varepsilon_{mix2}=\varepsilon_{f2}{}^{V_{f2}}\varepsilon_{a}{}^{V_{a}},\;$$
where $V_{f2}$ and $\varepsilon_{f2}$ have similar definitions to $V_{f1}$ and $\varepsilon_{f1}$. $V_{a}$ and $\varepsilon_{a}$ are the volume ratio and dielectric constant of the air and vacuum in Region 2. By definition,
(6)$$\varepsilon_{a}=1\;\;$$

Supposing that the properties of the fibers are uniform in both regions,
(7)$$\varepsilon_{f1}=\varepsilon_{f2}=\varepsilon_{f},\;$$
and
(8)$$V_{f1}=V_{f2}=V_{f}\;\;$$

In addition,
(9)$$V_{r}+V_{f}=V_{a}+V_{f}=1\;$$

Therefore,
(10)$$\varepsilon_{mix1}=\varepsilon_{f}{}^{V_{f}}\varepsilon_{r}{}^{1\;-V_{f}},\;$$
while
(11)$$\varepsilon_{mix2}={\;\varepsilon }_{f}{}^{V_{f}}.$$

As a result, the capacitances in Regions 1 and 2 are
(12)$$C_{1}=\frac{\varepsilon_{mix1}\varepsilon_{0}Wx}{d},\;$$
and
(13)$$C_{2}=\frac{\varepsilon_{mix2}\varepsilon_{0}W(L-x)}{d},\;$$
where W and  L are the width and length of the copper foil tape, respectively, and *x* is the position of flow front. According to the property of the parallel circuit, the total capacitance *C* is the sum of $C_{1}$ and $C_{2}$, i.e.,
(14)$${C=C}_{1}{+C}_{2}=\frac{\varepsilon_{mix1}\varepsilon_{0}Wx}{d}+\frac{\varepsilon_{mix2}\varepsilon_{0}W(L-x)}{d}.$$

Substituting (10) and (11) into (14), the total capacitance can be described by the following equation:(15)$$C=\frac{\varepsilon_{0}{W\varepsilon }_{f}{}^{V_{f}}\left(\varepsilon_{r}{}^{1\;-V_{f}}-1\right)}{d}x+\frac{\varepsilon_{0}{WL\varepsilon }_{f}{}^{V_{f}}}{d}.$$

In the above equation, the numbers *W*, *d*, and *L* are known constants, and $\varepsilon_{r}$ can be obtained by laboratory analysis. The value of *C* can be read from the capacitor in real time, while the flow front position *x* is available from the visualization system. The only unknown parameters in (15) are $V_{f}$ and $\varepsilon_{f}$, where $V_{f}$ may vary from across experiments and $\varepsilon_{f}$ is not easy to measure. By recording the changing values of *C* and *x* and fitting their relationship as a straight line, the unknown parameters $V_{f}$ and $\varepsilon_{f}$ can be solved from the slope and intercept of the line. Hence, the porosity of the fiber perform can be obtained as
(16)$$\phi =1-V_{f}\;$$

### 3.2. Darcy’s Law for Permeability Measurement

In the literature, Darcy’s law [20] is the core of most permeability measurement methods. The setup of Darcy’s experiments is illustrated in Figure 7, based on which it is found that the flow rate of the fluid through a porous medium is proportional to the cross-sectional area (A) and height ($h_{1}-h_{2}$) of the experimental apparatus yet inversely proportional to the flow distance (L).
(17)$$\;q=kA\frac{h_{1}-h_{2}}{L},\;$$
where q is the fluid flow rate, *k* is the permeability of the porous medium, A is the cross-sectional area, $h_{1}-h_{2}$ is the height, and *L* is the flow distance. A generalized version of the above equation is
(18)$$u=-\frac{K}{\mu }\cdot \nabla P$$
where K is the permeability tensor, ∇P is the pressure gradient, μ is the fluid viscosity, and u is the vector of Darcy velocity.

In the experiments designed in this work, only the in-plane permeability along the flow direction is focused on. Therefore, (18) can be simplified by making several common assumptions [21]: (1) The flow coordinate is in accordance with the principal fiber direction. (2) The flow of the fluid is one dimensional. (3) The depth direction is neglected.
(19)$$u=-\frac{K}{\mu }\left(\frac{\partial P}{\partial x}\right)\;$$
where K, $\frac{\partial P}{\partial x}$, and u are the medium permeability, pressure gradient, and Darcy velocity of the fluid along the flow coordinate, respectively. It is noted that the Darcy velocity cannot be visually observed. The CCD camera or other sensors used in the experiments only capture the seepage velocity at the flow front. The relationship between the seepage velocity $u^{{}^{\circ}}$ and the Darcy velocity u is as follows:(20)$$u=u^{{}^{\circ}}\phi \;$$
where ϕ is the porosity of the medium, i.e., the porosity of the fiber preform used in RTM, which can be obtained using the algorithm proposed in Section 3.1. 

In the literature, there are usually two types of infusion methods used for permeability measurements, one with constant pressure and the other with constant flow rate. In this work, the constant-pressure infusion method was adopted. The pressure at the inlet port was $P_{0\;}$ = 1 atm, while that at the outlet port was 0 atm because of the vacuum pumping. Therefore, it can be derived from (19) that
(21)$$x^{2}=\frac{2KP_{0}t}{\left(1-V_{f}\right)\mu }\;$$
where *t* is the infusion time and *x* is the location of the resin flow front at time *t*. In this equation, $V_{f}$ can be calculated using the method proposed in Section 3.1, while $P_{0}$ and μ are known constants. Therefore, by fitting a linear relationship between *t* and *x*^2^, the value of permeability (K) can be obtained from the slope.

It is notable that both ϕ and K can be obtained in a single experiment using the algorithms introduced in Section 3.1 and Section 3.2.

### 3.3. Numerical Simulation

After obtaining the material properties ϕ and K, numerical simulations can be conducted to verify the measurement accuracy. In detail, the measured parameters are input to the simulation software to calculate the flow phenomena of fluid in the mold cavity during the RTM process; then, the simulated flow front positions are compared with those recorded during the measurement experiments. In this work, Rhinoceros 3D version 5 (Rhino 5) was used to model the experimental equipment and achieve mesh generation, while Moldex3D was adopted for process simulation. The parameters used in numerical simulations are shown in Figure 8.

#### 3.3.1. Equipment Modeling and Mesh Generation

According to the mold structure used in the experiments, the mold cavity is modeled as a rectangular plate with dimensions 33 cm × 12 cm × 0.3 cm using Rhino 5. Then, the injection gate, vent location, and fabric layup orientation are set based on the actual conditions.

Consequently, the 2D surface mesh is generated, following by the 3D solid mesh generation by using the stretching function based on the 2D surface mesh. The final grid shows that the 3D mesh contains approximately 45,440 elements.

#### 3.3.2. Process Simulation

The resin is assumed to be incompressible. The non-isothermal 3D flow of resin in a porous preform is described by the following governing equations. The equation of continuity can be written as
(22)$$\frac{\partial \rho }{\partial t}+\nabla \cdot \rho u=0\;$$

Assuming that there is no density change in the experiments, the above equation reduces to
(23)∇⋅u=0 

The flow of resin in the fiber mats can be expressed by the Darcy’s law as expressed in (18). For analyzing the dynamic behavior of the resin flow, the permeability tensor ***K*** of the porous preform should be known,
(24)$$K=\left[\begin{array}{ccc} K_{xx} & K_{xy} & K_{xz}\\ K_{yx} & K_{yy} & K_{yz}\\ K_{zx} & K_{zy} & K_{zz} \end{array}\right]=\left[\begin{array}{ccc} l_{11} & l_{12} & l_{13}\\ l_{21} & l_{22} & l_{23}\\ l_{31} & l_{32} & l_{33} \end{array}\right]\left[\begin{array}{ccc} K_{11} & 0 & 0\\ 0 & K_{22} & 0\\ 0 & 0 & K_{33} \end{array}\right]\left[\begin{array}{ccc} l_{11} & l_{21} & l_{31}\\ l_{12} & l_{22} & l_{32}\\ l_{13} & l_{23} & l_{33} \end{array}\right]\;$$
where K*_ij_* (*i*, *j* = *x*, *y*, or *z*) are the components of the permeability tensor, $K_{11}$, $K_{22}$, and $K_{33}$ are the principal permeability, l*_ij_* are the directional cosines of the local coordinates, and
(25)$$\left[\begin{array}{ccc} l_{11} & l_{12} & l_{13}\\ l_{21} & l_{22} & l_{23}\\ l_{31} & l_{32} & l_{33} \end{array}\right]=\left[\begin{array}{ccc} \cos\alpha & -\sin\alpha & 0\\ \sin\alpha & \cos\alpha & 0\\ 0 & 0 & 1 \end{array}\right]\;$$

More details about these parameters can be found in the literature [22]. In the experiments conducted in this work, α was zero. Therefore, (24) reduces to
(26)$$K=\left[\begin{array}{ccc} K_{11} & 0 & 0\\ 0 & K_{22} & 0\\ 0 & 0 & K_{33} \end{array}\right]\;$$

In the situations of linear flow and thin mold cavity, only *K*_11_ dominates the resin flow behavior. Therefore, we set $K_{11}=K_{22}=K_{33}=K$ for simplicity.

The RTM process is simulated by using the Moldex3D software. The material parameters involved in the governing equations, such as fluid viscosity, and permeability and porosity of fiber reinforcement, are set, together with the injection pressure. Then, simulations are done by a finite volume method [23] based algorithm due to its robustness and efficiency. By comparing the flow fronts obtained from the simulation results and the real flow fronts recorded during the experiments, the accuracy of the permeability and porosity measurements can be verified.

## 4. Experimental Results

To illustrate the feasibility of the proposed method, experiments were conducted for measuring the porosity and permeability of two fiber preforms constructed with different layers of glass fiber sheets.

### 4.1. Experiments on Nine-Layer Fiber Preforms

The first case study was on nine-layer fiber performs, which were made of the glass fiber sheets shown in Figure 5. Figure 9 displays the images captured by the visualization system at different time points during one experiment. By reading the ruler, the changes of the flow front positions (*x*) were recorded. At the same time, the capacitance (*C*) was measured by the parallel-plate capacitor, as illustrated in Figure 10. Then, the relationship between *x* and *C* was fitted with a linear regression function. As shown in Figure 11, the regression between *x* and *C* can be expressed as
(27)C=3.8688x+58.118
in this case. By comparing (27) to (15), it was derived that
(28)$$\frac{{W\varepsilon }_{f}{}^{V_{f}}\left(\varepsilon_{r}{}^{1\;-V_{f}}-1\right)}{d}=\frac{3.8688}{\varepsilon_{0}}\;$$
and
(29)$$\frac{{WL\varepsilon }_{f}{}^{V_{f}}}{d}=\frac{58.118}{\varepsilon_{0}}.$$

After substituting other parameters into the above two equations, i.e., W = 1.5 cm, *d* = 0.3 cm, *L* = 30 cm, $\varepsilon_{r}=4.25$, and $\varepsilon_{0}=0.08855\;\mathrm{pF}/\mathrm{cm}$, it was calculated that $V_{f}$ = 0.242. Therefore, the porosity of the fiber preform used in this experiment is $\phi =1-V_{f}=0.758$. By substituting $V_{f}$ and other available parameters, i.e., *μ* = 560 cp and *P*_0_ = 1 atm, into the fitting result of (22), the permeability was estimated as $K=1.85\times 10^{-10}\;m^{2}$. Such results were consistent with our previous results achieved using another measurement system [13].

This experiment was repeated to ensure validity. As displayed in Figure 12 and Figure 13, the capacitance increased with the time and the flow front displacement. The relationship between *x* and *C* is
(30)C=3.9006x+57.584

Following a similar procedure to that introduced in the previous paragraph, it was calculated that $V_{f}$ = 0.233, $\phi =1-V_{f}=0.767$, and $K=2.07\times 10^{-10}\;m^{2}$. Such results are close to those obtained in the previous experiment.

### 4.2. Experiments on Seven-Layer Fiber Preforms

The fiber preforms used in the second case study were constructed with seven layers of glass fiber sheets. The fiber sheets used in this case study were purchased in a different batch from those used in the case study described in Section 4.1. Therefore, although the weave patterns of fibers were similar, there was still a certain degree of differences in material properties. The images captured by the visualization system are shown in Figure 14, from which the flow front displacements can be identified. The changes of capacitance are plotted in Figure 15. Combining the information recorded by both the visualization system and the capacitance sensor, the relationship between *x* and *C* can be estimated. As shown in Figure 16, the regression function is
(31)C=4.4579x+67.603 

From the slope and intercept terms, it can be calculated that $V_{f}$ = 0.246. Hence, $\phi =1-V_{f}=0.754$. According to (22) derived from the Darcy’s law, $K=4.26\times 10^{-10}\;m^{2}$.

In the repeated experiment, similar results were achieved. The capacitance values recorded at different time points are shown in Figure 17, while the relationship between *x* and *C* is plotted in Figure 18. Based on the regression function
(32)C=4.5036x+73.924 
it was derived that $V_{f}$ = 0.282, while $\phi =1-V_{f}=0.718$. Then, the permeability was calculated by using (22), which equaled 2.9 × 10^−10^ m^2^.

Comparing to the results of the first case study (Table 1), the porosity values measured in this case study (Table 2) show a reduction in reproducibility. As shown in some previous research [18,24], the porosity–permeability relationship is not linear; instead, it can be described as an exponential function. This is the reason why a small deviation in porosity causes a relatively large deviation in permeability.

### 4.3. Verifications with Simulations

The measurement results were then verified with numerical simulations as introduced in Section 3.3.2. According to the operation condition, the infusion pressure was 1 atm. The resin viscosity was set to 560 cp.

For simulating the resin flow in the experiments on the nine-layer fiber preforms, the average values of the estimated permeability and porosity were used as shown in Table 1. The comparison between the experimental and simulated flow fronts is displayed in Figure 19, which plots the flow front positions along time. A similar pattern between the simulation and experimental results can be observed. In addition, the snapshots at the 50th and 450th second after the infusion started are shown in Figure 20 and Figure 21 to further visualize the comparison, where the triangle and circle symbols represent the two experiments, and the solid curve represents the simulation results. The experimental and simulated flow fronts are quite close to each other.

The experiments on the seven-layer fiber preforms were verified in the similar way. The permeability and porosity were set according to the average values of the experimental results as shown in Table 2. The experimental and simulated flow front positions are plotted in Figure 22 along time. Again, the simulation results are fairly consistent with the recorded flow front positions. Figure 23 shows the snapshot at the 250th second. The deviations between the experimental and simulation results are larger than those obtained in the case study on the nine-layer fiber preforms but still acceptable, the reasons for this include possible distortion of the mold plates, local variations in material properties, variations in resin viscosity, etc.

## 5. Conclusions

In RTM, the permeability and porosity of fibers are important parameters that influence the resin flow properties. In this work, we used a parallel-plate capacitor to record the capacitance between the top and bottom mold plates. Although the parallel-plate capacitor is not a new technique, the algorithms developed in this paper with the aid of a visualization system make this conventional measurement device capable of measuring the permeability and porosity of the fiber preforms at the same time. In the designed experimental system, the parallel-plate capacitor only covers the center part of the fiber sheet. Accordingly, only the center positions of the flow front were measured for the porosity and permeability estimation. The center part of the resin flow is least affected by race-tracking; therefore, the information from it is suitable to be used for the estimation of material properties. The viscosity of the resin used in this work is about 560 cp. According to the literature [25], resin with viscosity smaller than 3000 cp should be acceptable for conducting the measurement experiments. The experimental results illustrate the feasibility of the proposed method. In this work, the accuracy of the measurements was further confirmed with numerical simulations.

In the end, we would like to point out the main limitations of the developed measurement system. In this system, the parallel-plate capacitor is installed inside the mold and in contact with the fibers. As a result, this system is not suited to the permeability measurement of conductive fibers, such as carbon fibers. In addition, the mold used in this work for illustrating the proposed method is not very thick, which may lead to undesired distortion. The possible distortion may change the porosity of the fiber reinforcement, which then affects the permeability. Fortunately, the method proposed in this work measures both porosity and permeability simultaneously, which means that the effects of mold distortion can be reflected by the measurement results. To achieve more-accurate measures of material parameters, a thicker mold is suggested.

## Figures and Tables

**Figure 1 polymers-14-02652-f001:**
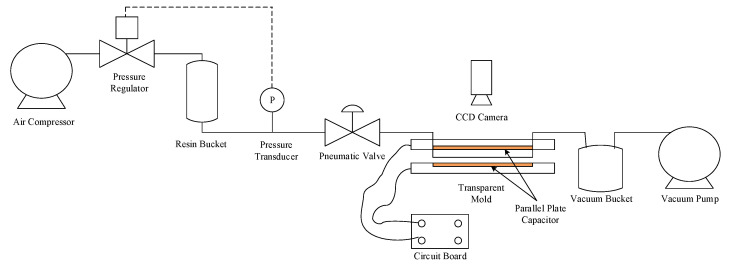
Process diagram.

**Figure 2 polymers-14-02652-f002:**
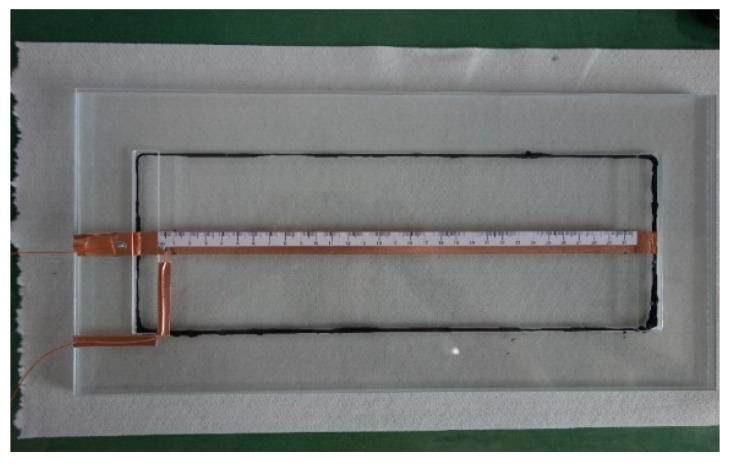
Transparent mold with parallel-plate capacitor.

**Figure 3 polymers-14-02652-f003:**
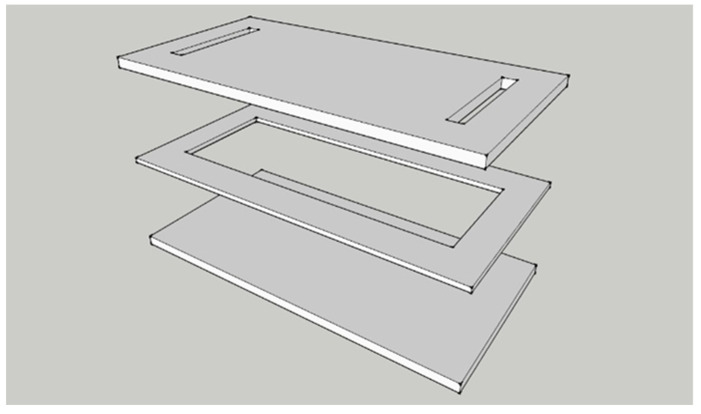
Conceptual illustration of transparent mold.

**Figure 4 polymers-14-02652-f004:**
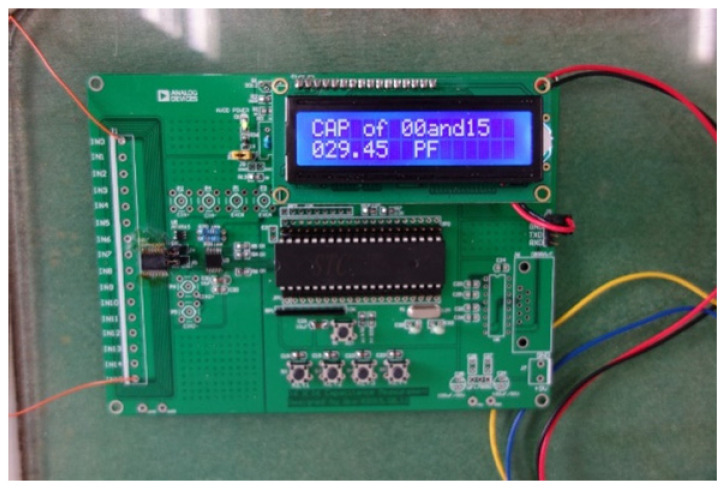
Circuit board for capacitance measurement and communication.

**Figure 5 polymers-14-02652-f005:**
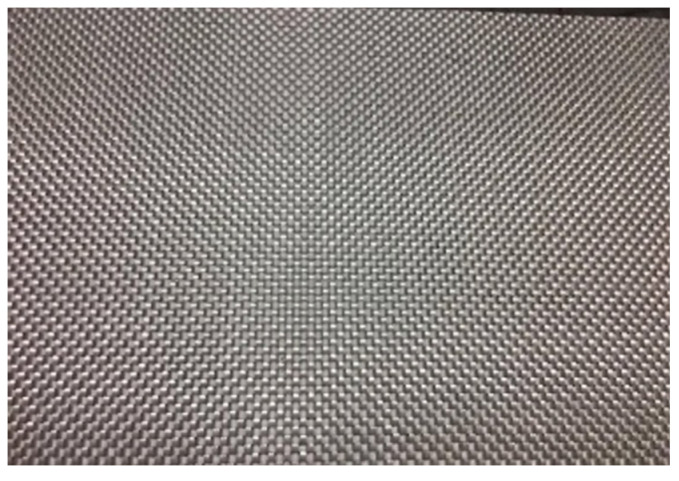
Glass fiber sheet.

**Figure 6 polymers-14-02652-f006:**
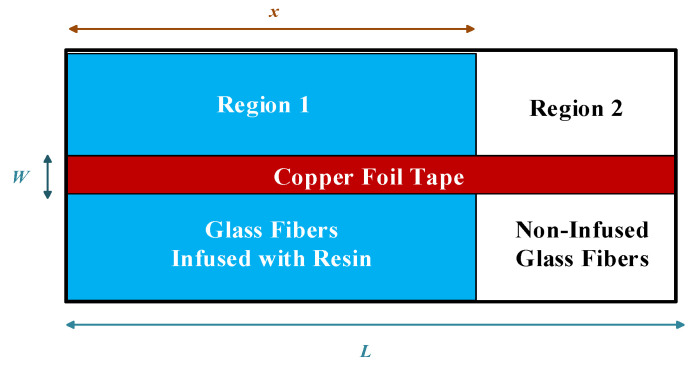
Illustration of top view of mold cavity.

**Figure 7 polymers-14-02652-f007:**
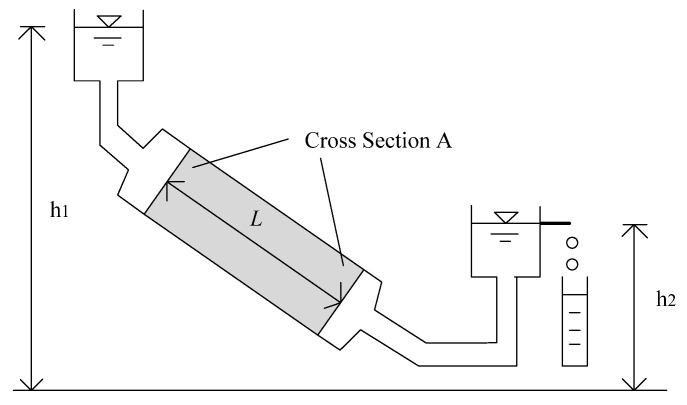
Illustration of Darcy’s law.

**Figure 8 polymers-14-02652-f008:**
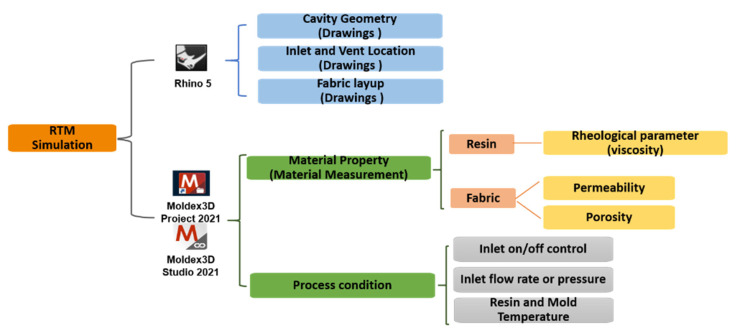
Parameters used in numerical simulations.

**Figure 9 polymers-14-02652-f009:**
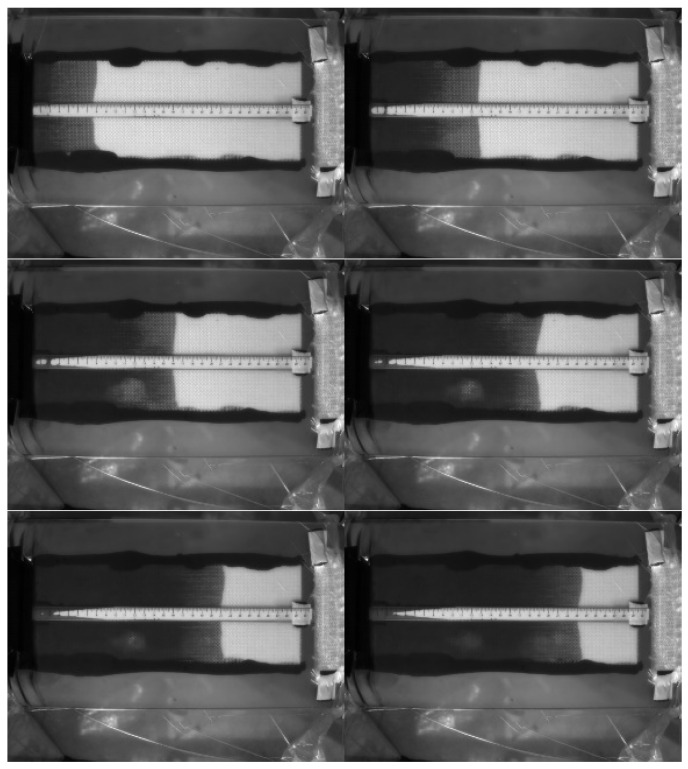
Flow visualization of RTM using a nine-layer fiber preform: images captured at the 50th, 150th, 250th, 350th, 450th, and 550th second.

**Figure 10 polymers-14-02652-f010:**
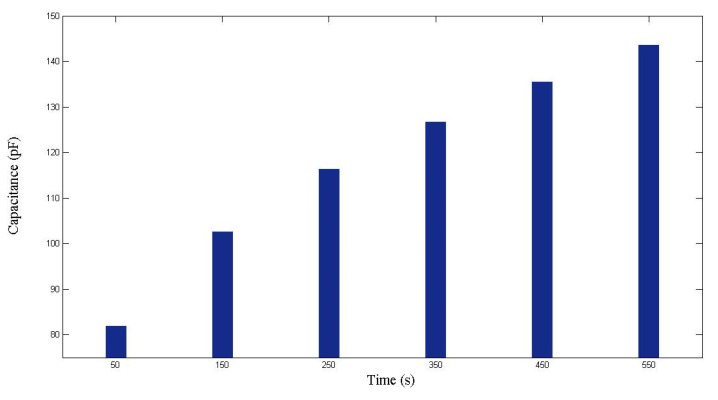
Capacitance measured during the first RTM experiment using a nine-layer fiber preform.

**Figure 11 polymers-14-02652-f011:**
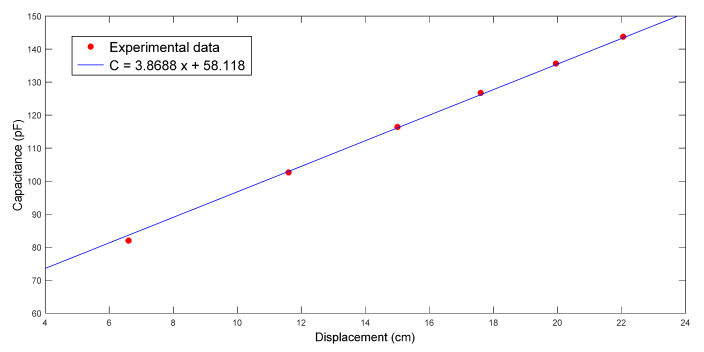
Relationship between flow front displacement and capacitance in the first RTM experiment using a nine-layer fiber preform.

**Figure 12 polymers-14-02652-f012:**
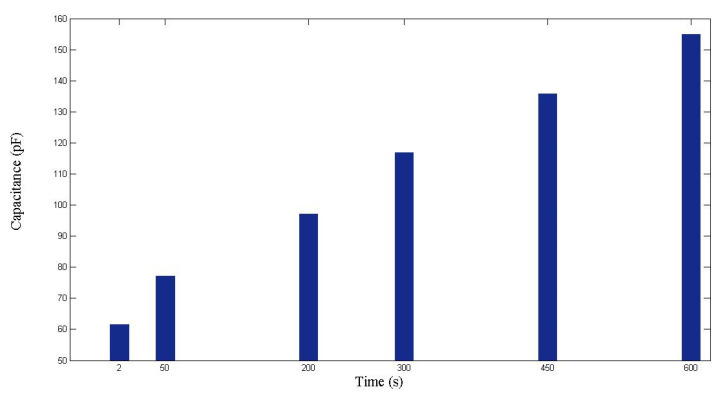
Capacitance measured during the second RTM experiment using a nine-layer fiber preform.

**Figure 13 polymers-14-02652-f013:**
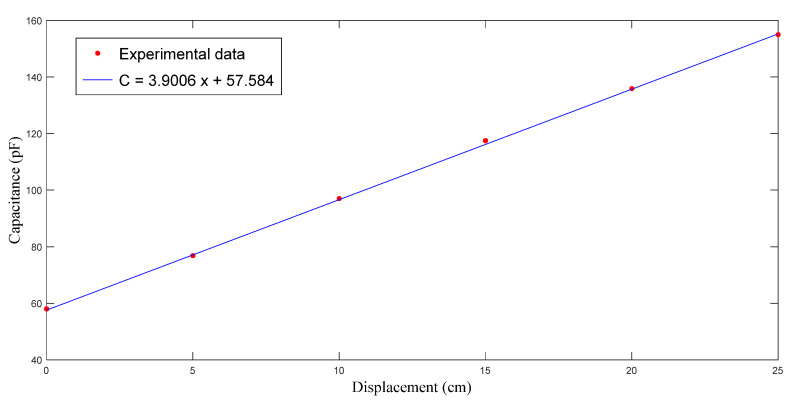
Relationship between flow front displacement and capacitance in the second RTM experiment using a nine-layer fiber preform.

**Figure 14 polymers-14-02652-f014:**
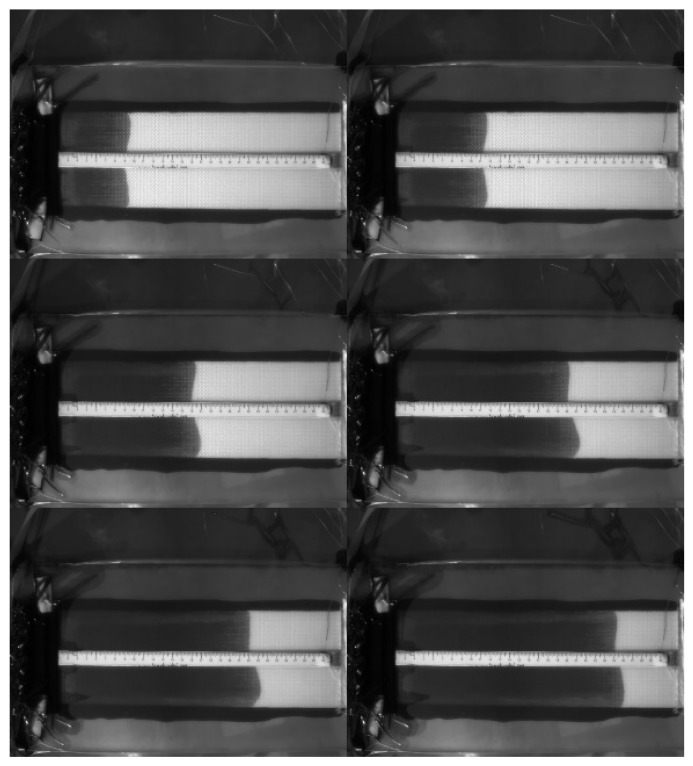
Flow visualization of RTM using a seven-layer fiber preform: images captured at the 50th, 100th, 150th, 200th, 250th, and 300th second.

**Figure 15 polymers-14-02652-f015:**
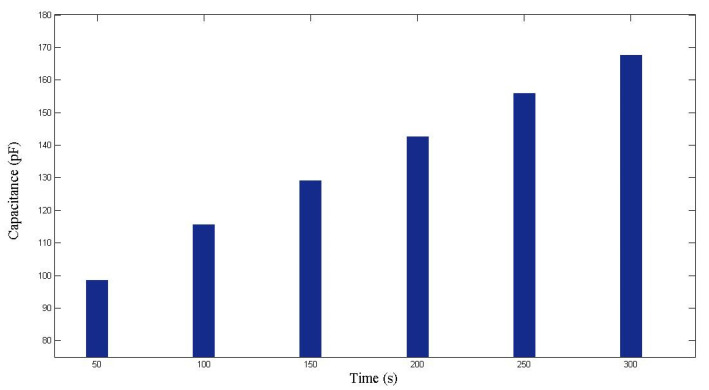
Capacitance measured during the first RTM experiment using a seven-layer fiber preform.

**Figure 16 polymers-14-02652-f016:**
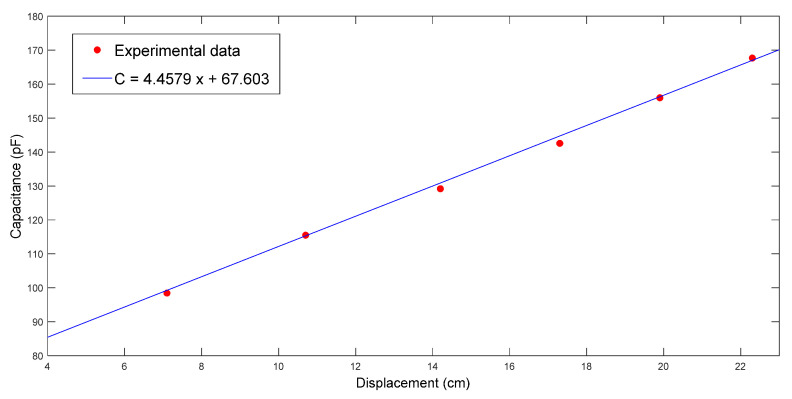
Relationship between flow front displacement and capacitance in the first RTM experiment using a seven-layer fiber preform.

**Figure 17 polymers-14-02652-f017:**
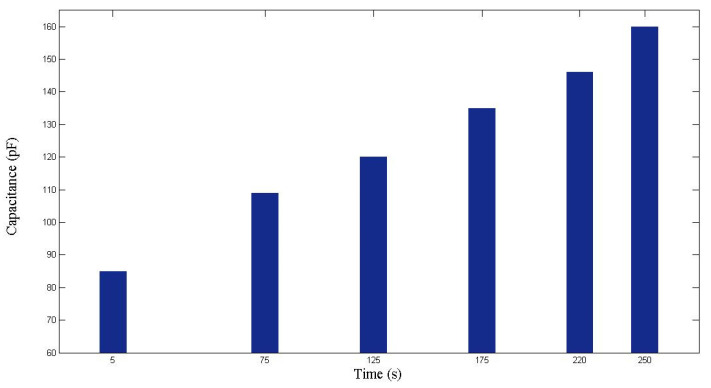
Capacitance measured during the second RTM experiment using a seven-layer fiber preform.

**Figure 18 polymers-14-02652-f018:**
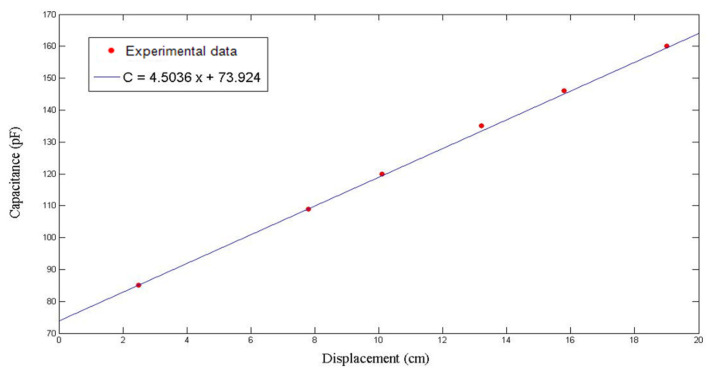
Relationship between flow front displacement and capacitance in the second RTM experiment using a seven-layer fiber preform.

**Figure 19 polymers-14-02652-f019:**
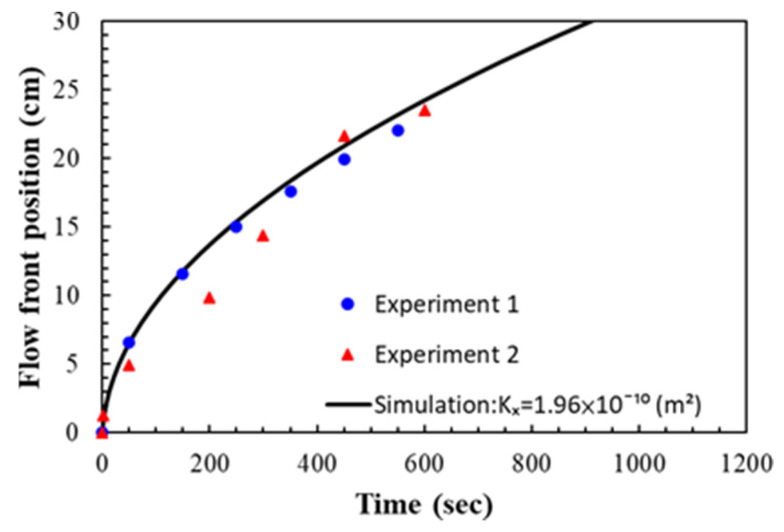
Comparison between experimental and simulated flow front positions in the case study of nine-layer fiber preform.

**Figure 20 polymers-14-02652-f020:**
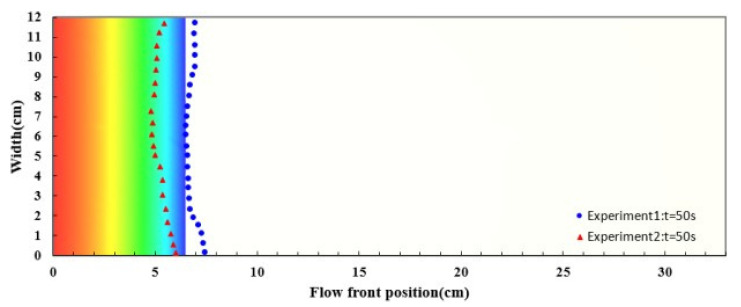
Snapshot of flow front positions at the 50th second in the case study of nine-layer fiber preform.

**Figure 21 polymers-14-02652-f021:**
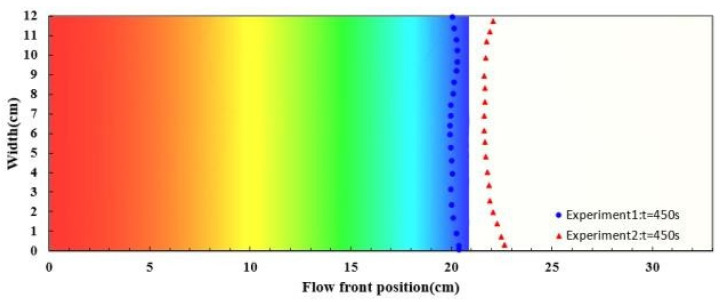
Snapshot of flow front positions at the 450th second in the case study of nine-layer fiber preform.

**Figure 22 polymers-14-02652-f022:**
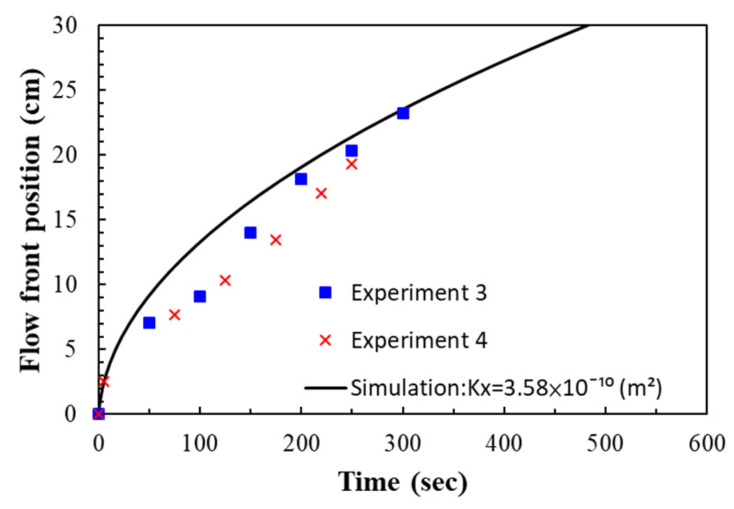
Comparison between experimental and simulated flow front positions in the case study of seven-layer fiber preform.

**Figure 23 polymers-14-02652-f023:**
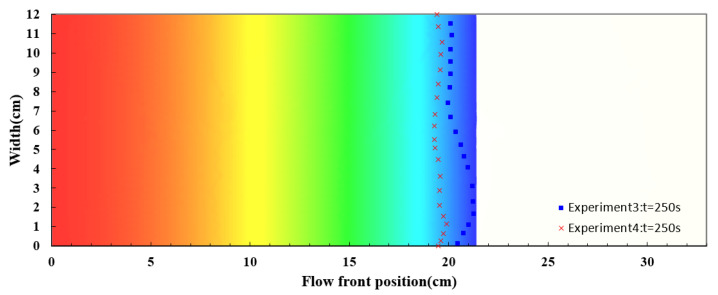
Snapshot of flow front positions at the 250th second in the case study of seven-layer fiber preform.

**Table 1 polymers-14-02652-t001:** Parameters of material property of the nine-layer fiber preform.

Parameters	Porosity	Permeability (m^2^)
Measured values from experiment 1	0.758	1.85 × 10^−10^
Measured values from experiment 2	0.767	2.07 × 10^−10^
Average values used in numerical simulation	0.763	1.96 × 10^−10^

**Table 2 polymers-14-02652-t002:** Parameters of material property of the seven-layer fiber preform.

Parameters	Porosity	Permeability (m^2^)
Measured values from experiment 1	0.754	4.26 × 10^−10^
Measured values from experiment 2	0.718	2.9 × 10^−10^
Average values used in numerical simulation	0.736	3.58 × 10^−10^

## Data Availability

The data presented in this study are available on request from the corresponding author.

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
