# Peer review of "Sensor Fusion for Simultaneous Estimation of In-Plane Permeability and Porosity of Fiber Reinforcement in Resin Transfer Molding"

_polymers, 2022, doi:10.3390/polym14132652_

Round 1

Reviewer 1 Report

1. As a 1D measurement, experimental details on how to ensure that the flow front is a straingt line? When the flow front is not straight, which is that case indicate in the manuscript, how to determine its acturate position?

2. The transparent plate mold is often used in in-plane permeability measurement. However, measures should be taken to avoid race tracking effect and to ensure a equilibrium and constant compaction of the fabric preforms. Especially the compaction status is important for the determination of the prosity and the permeability. This is quite challenging in VARTM process since the compaction of the fabric preform is changing with the resin infiltration. The fabric center is often more compacted than its periphery location. 

3. In table 2, why the measured permeability has such a huge deviation in two measurements, more than 30%?And the porosity seems to have very good reproduciability?The authors should provide more discussions on their results.

Reviewer 2 Report

This article is very interesting because it proposes a method for measuring fiber sheet parameters that is essential for RTM simulation. I recommend that the following points be corrected before publication.

1. Please comment on the mold distortion.

It is important that the height in the cavity is aligned for accurate measurement. When the total thickness is 0.9 cm and the height in the cavity is 0.3 cm, then the lid and bottom are 0.3 cm thick.

What is the degree of distortion of the lid and bottom when a vacuum is applied? Please calculate based on the physical properties of the materials and show that there is no problem.

2. Regarding the injection of resin.

In this measurement, it is important that the resin is injected at the same rate in the center and at the periphery. This can be rephrased as a rectangular shape of the resin injection area. The resin may be supposed to flow from the slit-shaped inlet to the center and the periphery at the same rate, so please describe the method used.

3. About the resin used.

There is almost no mention of the resin to be injected. Please discuss whether any resin with known physical properties is acceptable, or whether the viscosity of the resin should be within a certain range.

4. Experimental results.

Experiment 1 in Fig. 18 and the simulation are in good agreement. Fig. 21, which the authors say quite consistent, is not a good agreement (especially Experiment 4). Please discuss the reason of it.

Round 2

Reviewer 1 Report

It is suggested that the authors try to evaluated their method in a real RTM test appratus. 

Author Response

Thank you for the suggestion. We would like to respond to the comment as follows.

  1. The proposed method is for simultaneous measurement of both porosity and permeability of the fiber reinforcement used in RTM, which includes both equipment design and algorithm derivation. In detail, the measurement device includes a rectangular transparent mold with a slit-shaped inlet and a slit-shaped outlet. The shape of the inlet and outlet relieves the race tracking effect and ensures a 1-D linear flow of the resin inside the mold cavity. The transparency of the mold allows the record of the flow front positions along time. In addition, the installation of the parallel plate capacitors inside the mold cavity provides a basis for porosity estimation. With the help of the proposed algorithm, both the porosity and permeability can be calculated simultaneously. From the above discussion, it can be seen that the equipment and the algorithm are integrative and cannot be used separately. Therefore, the proposed method cannot be used in an arbitrary RTM apparatus.
  2. In industrial applications, there are three steps. First, the porosity and permeability of the fiber reinforcement can be measured using the proposed equipment and algorithm. Then, the measured parameters are used in computer-aided engineering simulations. These simulations often aim to find optimal mold design or operating conditions for specific industrial applications. In the third step, RTM experiments are conducted to verify the simulation results. Our work presented in this paper only focuses on the first step, while steps 2 and 3 are out of the scope of this paper.
  3. To avoid misunderstanding, the sentence "for industrial applications, a thicker mold is suggested" has been revised to "to achieve more accurate measures of material parameters, a thicker mold is suggested".